# Correlation of Myeloid-Derived Suppressor Cell Expansion with Upregulated Transposable Elements in Severe COVID-19 Unveiled in Single-Cell RNA Sequencing Reanalysis

**DOI:** 10.3390/biomedicines12020315

**Published:** 2024-01-29

**Authors:** Mitra Farahmandnejad, Pouria Mosaddeghi, Mohammadreza Dorvash, Amirhossein Sakhteman, Manica Negahdaripour, Pouya Faridi

**Affiliations:** 1Quality Control of Drug Products Department, School of Pharmacy, Shiraz University of Medical Sciences, Shiraz 71348-14336, Iran; Farahmandmitra93@gmail.com; 2Pharmaceutical Sciences Research Center, Shiraz University of Medical Sciences, Shiraz 71348-14336, Iran; 3Department of Pharmaceutical Biotechnology, School of Pharmacy, Shiraz University of Medical Sciences, Shiraz 71348-14336, Iran; 4Medicinal Plants Processing Research Center, School of Pharmacy, Shiraz University of Medical Science, Shiraz 71348-14336, Iran; pouriamosaddeghi@gmail.com; 5Student Research Committee, Shiraz University of Medical Sciences, Shiraz 71348-14336, Iran; 6Department of Biochemistry and Molecular Biology, Biomedicine Discovery Institute, Monash University, Clayton, VIC 3800, Australia; mohammadreza.dorvash@monash.edu; 7Proteomics and Bioanalytics, Department of Molecular Life Sciences, School of Life Sciences, Technical University of Munich, 80333 Munich, Germany; amirhossein.sakhteman@tum.de; 8Monash Proteomics and Metabolomics Platform, Department of Medicine, School of Clinical Sciences, Monash University, Clayton, VIC 3800, Australia; 9Centre for Cancer Research, Hudson Institute of Medical Research, Clayton, VIC 3168, Australia; 10Department of Molecular and Translational Science, Faculty of Medicine, Nursing and Health Sciences, Monash University, Clayton, VIC 3168, Australia

**Keywords:** COVID-19, single-cell RNA sequencing, transposable elements, immune system, myeloid-derived suppressor cells

## Abstract

Some studies have investigated the potential role of transposable elements (TEs) in COVID-19 pathogenesis and complications. However, to the best of our knowledge, there is no study to examine the possible association of TE expression in cell functions and its potential role in COVID-19 immune response at the single-cell level. In this study, we reanalyzed single-cell RNA seq data of bronchoalveolar lavage (BAL) samples obtained from six severe COVID-19 patients and three healthy donors to assess the probable correlation of TE expression with the immune responses induced by the severe acute respiratory syndrome coronavirus-2 (SARS-CoV-2) in COVID-19 patients. Our findings indicate that the expansion of myeloid-derived suppressor cells (MDSCs) may be a characteristic feature of COVID-19. Additionally, a significant increase in TE expression in MDSCs was observed. This upregulation of TEs in COVID-19 may be linked to the adaptability of these cells in response to their microenvironments. Furthermore, it appears that the identification of overexpressed TEs by pattern recognition receptors (PRRs) in MDSCs may enhance the suppressive capacity of these cells. Thus, this study emphasizes the crucial role of TEs in the functionality of MDSCs during COVID-19.

## 1. Introduction

The rise of severe acute respiratory syndrome coronavirus-2 (SARS-CoV-2) in late 2019 led to a massive crisis and a global pandemic. Since its inception, it has resulted in over 689 million reported cases and more than 6.8 million deaths “https://www.worldometers.info/coronavirus/ (accessed on 31 May 2023)”. COVID-19 exhibits a vast spectrum of clinical manifestations based on disease severity in different cases of infection, ranging from mild upper respiratory tract symptoms to severe, multi-organ failure, and even death. Although several studies have been carried out to unravel the detailed mechanistic understanding of COVID-19, the underlying pathogenesis of COVID-19 and its optimal treatment are still far from being clearly understood [1,2,3].

Transposable elements (TEs) make up around 50 percent of the human genome and are classified into retrotransposons (class I) and DNA transposons (class II) [4]. Retrotransposons, which amplify themselves in the genome, are classified into long terminal repeat (LTR) and non-LTR elements. LTR elements consist of around eight percent of the human genome and are categorized into four families: endogenous retrovirus (ERV), ERV-K, ERV-L, and MaRL. In contrast, non-LTR retrotransposons consist of long-interspersed elements (LINEs), short-interspersed elements (SINEs), and Sine-VNTR-Alu (SVA) [5]. It has been debated that the dysregulation of TE expression participates in several diseases, from cancer to neurological disorders [5,6,7,8,9,10].

Silencing TEs in hosts is crucial to prevent genome instability and inflammation [11] However, recent evidence indicates that during a viral infection, TEs may be reactivated and utilized by the host for antiviral defense. TE expression is a common occurrence in various species and cell types during viral infections. This up-regulation of TEs happens early in the infection process, even before significant increases in virus replication and interferon gene expression, and is observed near genes involved in antiviral defense and the response to interferons, suggesting a link to the host’s immune response. TE mRNAs and proteins have the potential to activate the innate immune response by triggering pattern recognition receptors (PRRs) [12].

Furthermore, the protein TRIM28, which plays a role in gene regulation and TE silencing, is found at higher levels in the promoter regions of several antiviral genes. When functional TRIM28 is lost, repression marks decrease at these promoters, leading to increased gene transcription. In the case of HIV-1, activation of specific TEs known as *LTR12C*, found upstream of interferon-inducible antiviral genes, has been shown to enhance antiviral defenses in a way that depends on the promoter. Therefore, the derepression of TEs near genes involved in the host’s innate immune system during infection is likely to have an immediate and significant impact on the host’s defenses [13].

Another study explored the role of TEs in the variability of individual responses to influenza A virus (IAV) infection. The data also revealed an inverse relationship between the basal transcripts of TEs and viral load after infection, suggesting that TE transcription contributes to the activation of innate immunity. Specific families of TEs were identified to be associated with changes in chromatin accessibility following infection. These families possess unique sequence characteristics, chromatin states, and an enrichment of binding motifs for transcription factors. This suggests that these TEs may influence the diverse responses individuals have to infection [14].

Recently, some studies have proposed the possible associations between COVID-19 severity and TEs [15,16,17,18,19,20]. Since TEs are activated and involved in inflammatory diseases, various studies suggest the potential role of TEs in COVID-19 pathogenesis. In this regard, Kitsou et al. revealed that TEs are strongly dysregulated in bronchoalveolar lavage fluid (BAL) samples of COVID-19 patients [16]. Moreover, Zhang et al. suggested that derepression of LINE expression, induced by SARS-CoV2 infection, and consequent inflammatory response, could cause SARS-CoV2 integration into the genome of infected cells [19].

Some studies have been carried out to decipher the COVID-19 pathology at the single-cell level [21,22,23]. However, there is no single-cell RNA seq study to examine the possible association of TE dysregulation in cell function and its potential role in COVID-19 immune dysregulation, to the best of our knowledge.

Herein, we reanalyzed RNA-seq data from the bronchoalveolar lavage (BAL) samples obtained from eight severe COVID-19 patients and four healthy donors at the single-cell level to investigate the potential relationship between TE expression and the immune responses triggered by the SARS-CoV-2 virus in COVID-19 patients.

## 2. Materials and Methods

The sc-RNA sequencing data of BAL samples from eight COVID-19 patients (including both alive and dead patients) and four healthy donors were retrieved from the gene expression omnibus GEO [24,25] database under accession numbers GSE157344 [21] and GSE151928 [26], respectively, and were extensively analyzed. In order to minimize the influence of potential comorbidities, we selected four individuals from each study group who were the youngest in age. The overall workflow of this method generated by https://biorender.com is depicted in Figure 1. Moreover, the code is available at https://github.com/mitra-frn/sc-TE-RNA/.

### 2.1. Aligning Reads

Patients’ paired-end FASTQ files were aligned to UCSC hg38 genome assembly using STARsolo (version 2.7.10a) [27] and v*3* (3M-february-2018.txt) cell *barcode whitelist files* with the setting’ –outSAMtype BAM SortedByCoordinate, –winAnchorMultimapNmax 100, –outFilterMultimapNmax 100, –outMultimapperOrder Random, –outSAMmultNmax 1.

The aligned sc-RNA sequence reads of healthy donor samples were directly obtained from the ENA database. They were aligned to hg38 genome assembly using STAR aligner as well.

### 2.2. Quantifying TE Expression

The traditional aligners cannot obtain an accurate quantitation of reads aligned to TEs, as they ignore multi-mapping reads, which is critical for counting TEs. Therefore, we used scTE (version 1.0.0) [28] aligner, which is compatible with STARsolo [27] output files.

Genome indices were built with the *scTE-build* function using UCSC genome browser Repeatmasker track [29] and GENCODE in two modes: exclusive and nointron, and BAM files were realigned via *scTE* function. Then, count matrices made by scTE were applied for further analysis.

### 2.3. Preprocessing, Analysis, and Exploration of scRNA-seq Data

The Seurat (version 3.2.3) [30,31,32,33] is an R packagesatijalab.org/Seurat) designed to analyze count matrixes and visualize data: At first, low-quality cells and cell doublets were filtered out based on these criteria: 2% of cells with greater read-count RNA, 2% of cells of each sample with low numbers of genes, or cells with more than 40% mitochondrial counts.

The filtered data were then normalized using ‘LogNormalize’ methods, and 2000 of most variable genes were determined using the ‘FindVariableFeatures’ function in Seurat. Samples were then integrated using the ‘IntegrateData’ function from the Seurat package to correct the batch effect.

### 2.4. Principal Component Analysis (PCA) and Clustering

The TEs were temporarily removed from the count matrix to ensure that they did not affect our clustering and were analyzed independently. After scaling the data, PCA was performed using the RunPCA function in Seurat with default parameters.

The K-Nearest Neighbors Algorithm (KNN) graph was conducted based on the PCA-reduced data, and unsupervised clustering was performed using ‘FindNeighbors’ and ‘FindClusters’. The TEs were added to clustered count matrix for further analysis.

### 2.5. Differential Expression (DE)

DE analysis was conducted on ‘RNA’ assays of count matrix, based on the ‘MAST’ [34] method using ‘FindAllMarkers,’ and cell type was determined using cell markers.

### 2.6. Scoring Pathways and Correlation Test

The gene sets of pathways that are presumably related to the immune system functions were retrieved from Gene Ontology (GO) [35,36], wiki pathway [37], Kyoto Encyclopedia of Genes and Genomes (KEGG) [38,39,40], BioPlanet [41], and Reactome [42] databases (Appendix A). Average expression levels of these gene set in myeloid clusters and upregulated genes (logFC > 0 and adj. *p*-value < 0.05) were calculated using ‘AddModuleScore’ in the Seurat package.

The correlation between these pathways and upregulated genes in MDSC clusters was evaluated using a calculated score with the ‘corr. test’ function in R version 4.0.4.

## 3. Results

To decipher the potential role of TE dysregulation in the immune deviation induced by the SARS-CoV-2 virus in COVID-19 patients, scRNA-seq data, including a total of 324,135 cells in the BAL samples obtained from eight severe patients and four healthy donors were reanalyzed (Table 1) [21,26].

Using a KNN graph constructed on the PCA-reduced data, single cells were clustered and then labeled based on their canonical cell markers (Appendix A). Average and percent expression of canonical cell markers were shown in all clusters of three populations (healthy, alive, and dead patients with severe COVID-19) (Appendix A). Also, to visualize these clusters, uniform manifold approximation and projection (UMAP) plots in these three populations are shown below (Figure 2).

Based on the canonical markers, we identified major cell types, including neutrophils, myeloid-derived suppressor cells (MDSCs), monocytes and macrophages, dendritic cells, lymphoid cells, and epithelial cells. Notably, among these cell types, MDSCs, specifically cluster 11 in the healthy population, clusters 7 and 8 in the alive population, and cluster 4 in the dead population, exhibited a distinct pattern of transcriptional element (TE) upregulation (Appendix A). These clusters demonstrated differential expression of a significant number of TEs (logFC > 0 and adj. *p*-value < 0.05). As illustrated in Figure 3, MDSC clusters exhibited a higher number of upregulated TEs compared to other myeloid cell-type clusters. Furthermore, MDSC clusters exhibited a higher frequency in infected patients. While MDSC clusters accounted for only 2% of all cells in the healthy population, they represented 10% and 7% of the cell population in the alive and dead populations, respectively.

### MDSCs Reveal Distinct Immunological Functions

MDSC clusters were identified in all healthy, survived, and dead patients, as labeled in Figure 2.

By scoring each pathway based on the expression levels of its genes, the correlation between the overexpressed genes in myeloid cell-type clusters (LogFC > 0 and adj. *p*-value < 0.05) and the selected pathways was evaluated for the healthy, alive, and dead populations. The results of this analysis are presented in Appendix A.

As demonstrated in Appendix A and Appendix A, there is a significant difference in the function of MDSCs and other myeloid cell-type clusters. To facilitate a comparison, we present seven pathways that exhibit greater variation among the first 12 myeloid cell clusters as they represent the major population of myeloid cells. Notably, certain pathways associated with neutrophil function displayed a negative correlation with MDSC function, indicating the suppressive nature of these cells. Conversely, pathways such as Toll receptor signaling, TNF signaling, and NF-kappa B signaling exhibited a positive association with MDSC function (Figure 4).

## 4. Discussion

This study aimed to examine whether TE dysregulation correlates with the immune deviation induced by the SARS-CoV-2 virus in COVID-19 patients. To answer this question, the BAL sample obtained from 12 samples (including 4 control, 4 alive patients, and 4 dead patients) were reanalyzed at the single-cell level. We identified a cluster of MDSC with a higher population in COVID-19 than the control.

Under normal homeostatic conditions, myeloid progenitors are produced in the bone marrow and appear as non-polarized or ‘resting’ MDSCs with very low suppressive activity. These cells can migrate to the periphery and differentiate into mature macrophages, dendritic cells, and neutrophils while losing their suppressive properties. However, under conditions of chronic inflammation, MDSCs including polymorphonuclear (PMN-MDSC) and monocytic (M-MDSC) ones undergo differentiation arrest, proliferation, and polarization toward highly suppressive cells that migrate to the periphery and sites of inflammation. Different subpopulations of MDSCs can sense changes in their surroundings and adjust their behavior accordingly. Therefore, in the context of chronic inflammation, the various subsets of MDSCs can sense changes in their environment and adapt accordingly due to their plastic nature. These cells can modify their developmental pattern, phenotype, and behavior in response to changes in environmental factors associated with chronic inflammation [43].

Research has shown that MDSCs play a critical role in regulating immune responses in various human diseases. MDSCs have the ability to inhibit T-cell proliferation and activation, modulate cytokine production by macrophages, suppress natural killer (NK) cell function, impair dendritic cell differentiation, and induce regulatory T cells (Tregs). Additionally, MDSCs can inhibit the proliferation and differentiation of B cells and induce regulatory B cells in multiple pathological conditions. These findings underscore the importance of MDSCs in the pathogenesis of diseases and highlight their potential as a therapeutic target [44,45,46].

Various studies have utilized advanced phenotypic and molecular techniques to investigate how the immune system interacts with the virus in COVID-19. Severe COVID-19 cases are characterized by alteration in the abundance, phenotype, and functionality of neutrophils. High numbers of neutrophils have been found in the nasopharyngeal epithelium, lungs, and blood of infected patients. Single-cell RNA sequencing has revealed the emergence of immature neutrophils that resemble PMN-MDSCs, suggesting the presence of immunosuppressive neutrophil precursors in severe COVID-19 cases. These precursors may be released prematurely from the bone marrow and infiltrate the lung tissue in severe cases, leading to the expansion of PMN-MDSCs and contributing to the observed neutrophilia [47,48].

We identified that TEs are highly upregulated in some clusters with a resemblance to myeloid cells. Notably, these cells were not recognized by canonical myeloid cell markers. These clusters were annotated as myeloid-derived suppressor cells based on the upregulation of some markers including Colony Stimulating Factor 3 Receptor (*CSF3R*) [49,50], Nicotinamide Phosphoribosyl transferase (*NAMPT*) [51,52], and nuclear paraspeckle assembly transcript 1 (*NEAT1*) [53,54]. The suppressive nature of MDSCs is in accordance with the obtained results suggestive of the negative association of canonical neutrophil functions including neutrophil migration, neutrophil-mediated immunity, neutrophil degranulation, and neutrophil activation involved in immune response with the MDSC function [55].

Long non-coding RNAs (lncRNAs), including *NEAT1* and metastasis-associated lung adenocarcinoma transcript 1 (*MALAT1*), were found to be highly upregulated in MDCS clusters with overexpressed TEs (Appendix A). This result is confirmed by others who demonstrated that most lncRNAs are expressed under the control of TE promoters [56]. Recent studies suggested that the upregulation of *NEAT1* and *MALAT1* may be associated with inflammation and consequent tissue damage seen in severe COVID-19 [57,58,59]. Moreover, it has been reported that TE overexpression is associated with inflammatory diseases. For instance, Macchietto et al. proved that TE overexpression is common in different viral infections [12]. The overexpression of TEs in SARS-CoV-2 infection is confirmed as well [16]. Noteworthy, autoantibodies against the endonuclease domain of the LINE1 gene are discovered in approximately 40% of SARS patients, which may be crucial in its pathogenesis [60].

Transcriptional regulatory networks are responsible for determining cellular identity, function, and response to stimuli by controlling gene expression programs. Regulatory elements such as promoters and enhancers act as ‘wires’ in the genome, connecting genes into regulatory networks and regulating nearby gene expression [61]. TEs have been suggested as playing a role in the evolution of regulatory networks due to their ability to replicate throughout the host genome. While most TEs no longer encode functional proteins, they often retain transcription factor binding sites and can influence the expression of nearby genes. In recent years, there have been several studies demonstrating the important role of TEs in host gene regulation, leading to the conclusion that the cooption of TEs is a general mechanism shaping the evolution of mammalian gene regulatory networks [62,63,64].

Recent research suggests that TEs are frequently coopted to regulate genes involved in immune processes. Studies have identified TEs being used as promoters [65], interferon-inducible enhancers [64], and insulator elements in immune cells [66]. The unique pressures involved in the evolution of immune regulatory networks may favor the cooption of TEs. Immune genes are among the fastest evolving genes in the genome, reflecting the constant need to adapt to new and evolving pathogens [61,64]. TEs, which are a major source of genetic polymorphism, may facilitate the rapid adaptive evolution of immune responses at the gene regulatory level due to their active or recently active status. In this regard, in a study by Ye et al. [4], chromatin profiling data from mouse CD8+ T lymphocytes was analyzed, revealing that multiple TE families contribute to predicted regulatory sequences. They also found that immune cells exhibit the highest enrichment of TE-derived enhancers compared to other cells, indicating that the cooption of TEs may more strongly influence immune regulatory networks. So, one may speculate that TE upregulation in MDSCs appears to be linked to the cells’ ability to adapt their phenotype and functional capabilities in response to changes in their microenvironment [43]. TEs are repetitive and mobile elements, and their involvement in the genomic plasticity of MDSCs may occur through their insertion into coding or regulatory regions of immune-related genes. This can have a functional impact on gene expression, leading to plasticity [11,61,67,68].

The defense against pathogens involves a series of coordinated events, beginning with the host’s recognition of the invading pathogen, in which Toll-like receptors (TLRs) play a crucial role. The purpose of sensing pathogens via TLRs is to quickly activate an innate immune response to eliminate the pathogen. Alveolar macrophages and neutrophils are well-known immune cells that are capable of phagocytosis and killing pathogens. Alveolar macrophages are usually the first responders, but they are later replaced by neutrophils, which are quickly recruited to the site of infection with the help of chemokines that are primarily produced by lung epithelial cells and macrophages. Neutrophils produce various harmful products, including reactive oxygen species and proteases, which can damage not only the pathogen but also the host’s own cells. After the pathogen has been eliminated, the host’s next priority is to initiate an appropriate anti-inflammatory response to prevent further neutrophil recruitment. Neutrophils have a short lifespan and begin to undergo apoptosis at the site of inflammation. To prevent lung injury, phagocytes must rapidly clear these apoptotic cells, a process known as efferocytosis. This is the stage at which MDSCs become involved in the host’s response to infection. MDSCs do not accumulate rapidly in the lung after infection. Instead, they develop late after infection, which makes sense because lung MDSCs produce IL-10 that can impede neutrophil recruitment if produced too early. It has been revealed that lung MDSCs efficiently clear apoptotic neutrophils through efferocytosis, aided by the IL-10 produced by the MDSCs. Successful pathogen clearance, reduction in neutrophil infiltration, and elimination of dead neutrophils ultimately restore tissue homeostasis. As we revealed in Figure 4, pathways related to neutrophil function were negatively associated with the function of MDSCs, which suggests these cells play an essential role in resolving lung inflammation by the inhibition of neutrophil functions [55].

The TLR2 and TLR4 signaling pathways induce the suppressive activity of MDSCs. Activation of the NF-κB pathway by TLR2/4 leads to the expression of inflammatory factors such as IL-6 and TNF-α. Subsequently, IL-6 and TNF-α activate both the STAT3 and NF-κB signaling pathways. Of particular note, the expression of the inflammatory factors S100A8 and S100A9 is regulated by STAT3. These factors act as TLR4 ligands, which then activate the NF-κB pathway, leading to an upregulation of IL-6 and TNF-α expression. This forms a feedback loop that enhances the expansion and activation of MDSCs [69].

As depicted in Figure 4, some pathways including the Toll receptor signaling pathway, TNF signaling pathway, and NF-kappa B signaling pathway were positively associated with the MDSCs function. Since PRRs, the same as TLRs, could recognize TEs [70,71], one may speculate that the upregulated TEs in MDSCs could enhance the suppressive activity of these cells. Although this assumption is based on limited evidence and requires further investigation to establish its validity [72], it serves as an initial proposition for further inquiry.

It should be noted that there were some limitations in our study. First of all, the number of samples that were reanalyzed was limited, and the lack of samples from mild patients in our analysis prevented us from generalizing this finding to COVID-19-mild patients. Moreover, in this study, autonomous TE expression was not distinguished from co-transcription or pervasive transcription. This might lead to overestimating TE expression and its functional effects on the studied process [73]. Another limitation of our study is that we did not thoroughly investigate the potential role of TE downregulation in gene expression related to COVID-19 infection. Our study observed a general upregulation of TEs, with fewer TEs being downregulated, thus excluding the role of TE downregulation in the gene expression changes observed. However, it is important to explore the possible impact of TE downregulation on gene expression in future studies to fully understand the complex dynamics between TEs and viral infections, particularly in the context of COVID-19.

All in all, this is the first study to examine the possible association of TE and its potential role in the COVID-19 immune responses at the single-cell level. Our results suggest that the expansion of MDSCs could be a hallmark of COVID-19. Moreover, we recognized that TEs are highly upregulated in MDSCs. The upregulation of TEs in COVID-19 could be related to the plasticity of these cells in response to the microenvironments. Moreover, it seems that the recognition of overexpressed TEs by PRRs in MDSCs could strengthen the suppressive activity of these cells. Therefore, this study underscores the importance of TEs in the functionality of MDSCs in COVID-19. However, further studies are needed to decipher the probable causal link between TE overexpression in MDSCs and their function seen in mild and severe COVID-19.

## Figures and Tables

**Figure 1 biomedicines-12-00315-f001:**
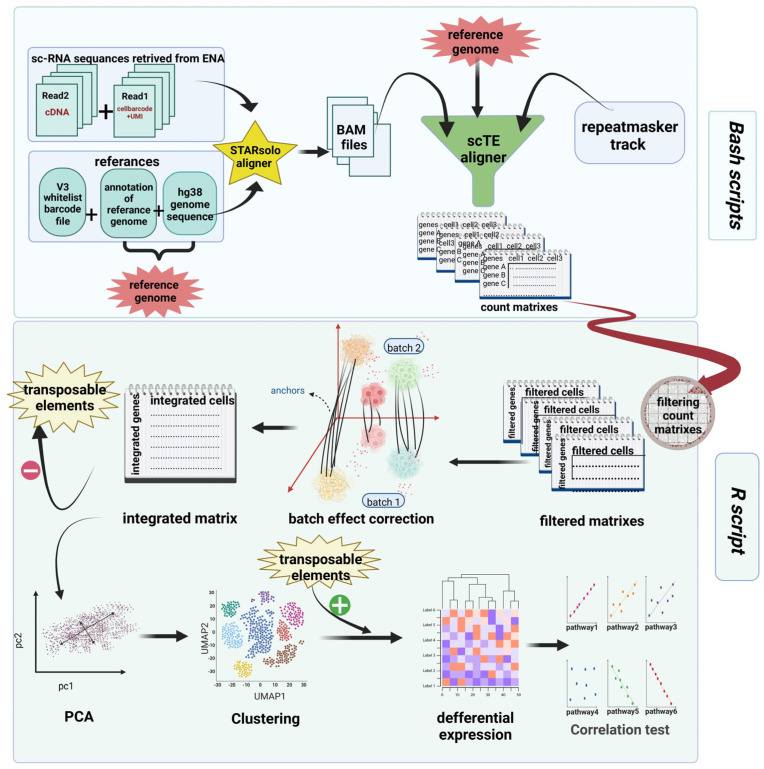
The workflow of this study. ScRNA reads were aligned to ‘hg38 reference genome’ and ‘repeatmasker tracks’ using STAR and sc–TE aligners to count both genes and TEs. In the next step, count matrixes were filtered, batch effects were removed, and all matrixes were integrated. Then, TEs were removed temporarily from the integrated matrix. After PCA and clustering, TEs were added, and DE analysis was conducted between all clusters. Finally, the correlation between DE TEs and some pathways was investigated. This figure was created by BioRender.

**Figure 2 biomedicines-12-00315-f002:**
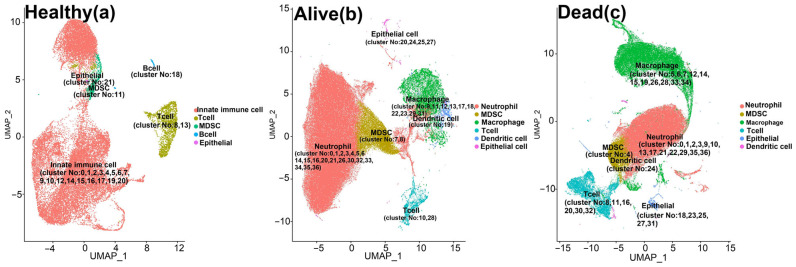
Single-cell analysis of cells from bronchoalveolar lavage (BAL) samples of individuals with COVID-19 and controls. This plot, which was obtained by UMAP visualization, demonstrates major cell types in BAL samples obtained from (**a**) healthy, (**b**) survived, and (**c**) dead patients. Each cell type is depicted by a specific color, and their respective clusters are indicated by cluster No.

**Figure 3 biomedicines-12-00315-f003:**
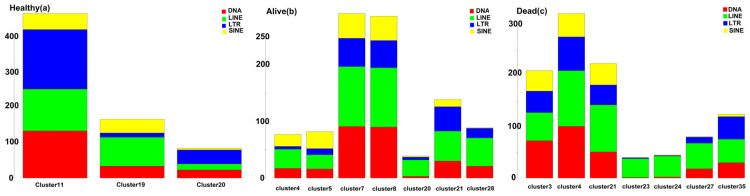
Bar plot summarizing the number of significantly upregulated transposable element (TE) subfamilies (logFC > 0 and adj. *p*-value < 0.05) in myeloid cell-type clusters from healthy samples (**a**), alive (**b**), and dead (**c**) patients. The subfamilies of transposable elements include DNA transposons, long-terminal repeat elements (LTR), long-interspersed elements (LINEs), and short-interspersed elements (SINEs) are depicted in red, blue, green, and yellow colors, respectively. The length of each bar represents the number of upregulated TEs from each subfamily. Clusters that have a TE number less than 50 are not represented in this illustration.

**Figure 4 biomedicines-12-00315-f004:**
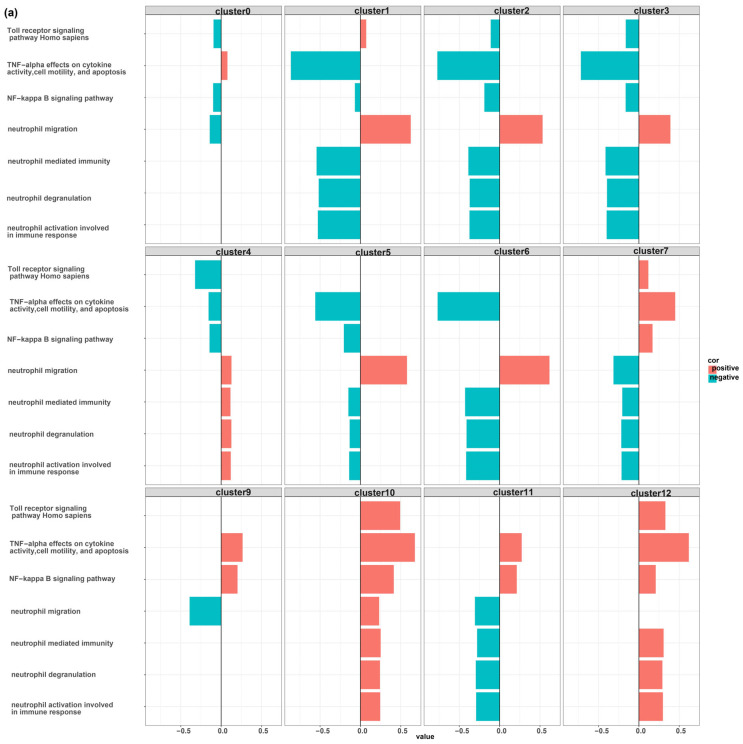
The correlation between seven specific pathways and overexpressed genes in the first 12 myeloid cell clusters from healthy, alive, and dead populations. There is a significant difference in the function of MDSC clusters with highly upregulated TEs compared to other myeloid cell types. MDSC clusters were cluster 11 of healthy samples (**a**), clusters 7 and 8 of alive patients (**b**), and cluster 4 of dead patients (**c**). MDSC clusters showed a negative correlation with neutrophil functions such as degranulation, activation involved in immune response, immunity, and migration. However, some pathways, including the Toll receptor signaling pathway, TNF signaling pathway, and NF-kappa B signaling pathway, were positively associated with the function of MDSCs.

**Table 1 biomedicines-12-00315-t001:** GEO accession number, clinical status, and clinical outcome of samples reanalyzed.

GEO Accession	Clinical Outcome	Clinical Status	Tissue
GSM4593888	-	Healthy	BAL
GSM4593891	-	Healthy	BAL
GSM4593890	-	Healthy	BAL
GSM4593892	-	Healthy	BAL
GSM4762143	Alive	Severe COVID	BAL
GSM4762155	Alive	Severe COVID	BAL
GSM4762159	Alive	Severe COVID	BAL
GSM4762144	Alive	Severe COVID	BAL
GSM4762140	Dead	Severe COVID	BAL
GSM4762152	Dead	Severe COVID	BAL
GSM4762150	Dead	Severe COVID	BAL
GSM4762147	Dead	Severe COVID	BAL

## Data Availability

The datasets reanalyzed for this study were retrieved from the gene expression omnibus GEO database under accession numbers GSE157344 and GSE151928, respectively. Moreover, the code used in this study is available at https://github.com/mitra-frn/sc-TE-RNA/.

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
