# Peer review of "Correlation of Myeloid-Derived Suppressor Cell Expansion with Upregulated Transposable Elements in Severe COVID-19 Unveiled in Single-Cell RNA Sequencing Reanalysis"

_biomedicines, 2024, doi:10.3390/biomedicines12020315_

Round 1

Reviewer 1 Report

Comments and Suggestions for Authors

The github page cannot be accessed. Please correct the link.

The Introduction is weak. Please expand on the role of TEs on general virus infection.

Figure 3 is hard to read. Please re-structure this figure to make it more easily interpretable. This may be because of the colors used.

Figure 4 is hard to read. Please re-structure this figure to make it more easily interpretable.

Author Response

The GitHub page cannot be accessed. Please correct the link.

We apologize for the inconvenience caused by the inaccessible GitHub link. We have taken note of the issue and corrected the link in the manuscript to ensure that it directs readers to the correct GitHub page. Thank you for bringing this to our attention, and we appreciate your understanding.

The Introduction is weak. Please expand on the role of TEs on general virus infection.

We appreciate the comment regarding the weakness of the Introduction section. In response, we have expanded the discussion on the role of Transposable Elements (TEs) in viral infections, providing a more comprehensive overview of their impact on the host's immune response.

Figure 3 is hard to read. Please re-structure this figure to make it more easily interpretable. This may be because of the colors used.

Thank you for your feedback regarding Figure 3. In response to your comment, we have restructured the figure to enhance its readability. This includes making improvements to the resolution . Additionally, we have expanded the caption to provide more detailed information about the figure's content and interpretation.

Figure 4 is hard to read. Please re-structure this figure to make it more easily interpretable.

Based on your feedback, we have made changes to the figure to improve its readability. These modifications involve enhancing the resolution and expanding the caption to offer more comprehensive details on the figure's content and its interpretation.

Reviewer 2 Report

Comments and Suggestions for Authors

It was a good idea to study potential TE function in COVID-19 pathogenesis, using scRNAseq data available in public database. However, the figures and results were not depicted well, making the manuscript hard to understand or difficult to believe.  

Major concerns:

1) Figure 2. an integrated clustering figure is appreciated, with separation for sample types (healthy, survived and dead) and with the same color assigned to the cell type in each. The current figures are biased, difficult to read and compare. The cell type annotations are also different from the context.

2) Figure 3. while only 6 major clusters are annotated in the context, these figures demonstrated various clusters which are not interpreted well, in either the manuscript or the supplementary materials. What are these clusters? What are their relationship with MDSC?

3) Figure 4. the clusters in these figures are totally different from those in Figures 2 and 3. Again, there is no annotation or interpretation at all regarding these cell types.

4) Mainly, from the manuscript, it is difficult for readers to figure out the correlation of TE with any cell type(s) relevant to COVID-19. When authors claimed "MDSC clusters are identified in survived and dead patients, as labeled in Figure 2" (Lines 188-189), I see MDSC cluster (only one, not clusters) in healthy as well. Nevertheless, the authors later studied the correlation in neutrophil clusters (Lines 134-135) without any reason given.

5) The authors only included up-regulated genes in analysis, excluding down-regulated genes, without any given reason.

Minor concerns:

1) Figure 1, "batch effect correction", not "btach effort correction".

2) Lines 65-66: "bronchoalveolar lavage (BAL) fluid", not BALF, since BALF is not used any more in the context. Using BAL hereafter.

3) Lines 90-93: same line for the same setting. 

4) Line 120: full name of KNN needs to be given.

5) Lines 138-143: legend for Figure 1, not a paragraph of context.

6) Line 145: "This" should be deleted.

7) ID is missing in most of GO, Wiki, KEGG or reactome analysis. GO should have been noted as CC, MF or BP. 

8) While the Discussion part is long, the Results part should have been more detailed for interpretation of figures and tables. 

Comments on the Quality of English Language

Please see above.

Author Response

Major concerns:

  • Figure 2. an integrated clustering figure is appreciated, with separation for sample types (healthy, survived and dead) and with the same color assigned to the cell type in each. The current figures are biased, difficult to read and compare. The cell type annotations are also different from the context.

Thank you for your feedback regarding Figure 2. In response to your comment, we have assigned each cluster with its relevant cell type, ensuring that certain clusters are associated with specific cell types. These adjustments aim to improve the overall readability and interpretability of Figure 2, allowing for a clearer understanding of the cell type distribution within the different sample types.

  • Figure 3. While only 6 major clusters are annotated in the context, these figures demonstrated various clusters which are not interpreted well, in either the manuscript or the supplementary materials. What are these clusters? What are their relationship with MDSC?

We apologize for any confusion caused by the interpretation of Figure 3. In response to your question, the clusters depicted in Figure 3 represent distinct subpopulations of myeloid cells. The manuscript discusses the relationship between these clusters and MDSCs (Myeloid-Derived Suppressor Cells), highlighting specific clusters associated with MDSCs. The bar plot in Figure 3 summarizes the number of significantly upregulated transposable element (TE) subfamilies (with logFC>0 and adj.p.value<0.05) within each myeloid cell type cluster from healthy samples (a), alive patients (b), and deceased patients (c). Notably, the number of upregulated TEs in MDSCs is considerably higher compared to other myeloid cell types.

  • Figure 4. the clusters in these figures are totally different from those in Figures 2 and 3. Again, there is no annotation or interpretation at all regarding these cell types.

Thank you for bringing this to our attention. We apologize for any confusion caused by the lack of annotations and interpretation regarding the cell types in the figures. We appreciate your understanding. Based on your feedback, we have made revisions to Figure 2 and the caption of Figure 3 to provide clearer clarification. In Figure 2, we have now assigned the clusters to their corresponding cell types, which improves the understanding of the cell type distribution. Additionally, in Figure 3, we have provided more detailed information about the upregulation of Transposable Elements (TEs) within the myeloid cell clusters. Myeloid clusters that have a TE number less than 50 are not represented in this illustration.

In Figure 4, we have conducted an enrichment analysis to explore the functional characteristics of the first 12 myeloid clusters, as they represent the major population of myeloid cells. These revisions aim to present a more comprehensive and interpretable representation of the data.

  • Mainly, from the manuscript, it is difficult for readers to figure out the correlation of TE with any cell type(s) relevant to COVID-19. When authors claimed "MDSC clusters are identified in survived and dead patients, as labeled in Figure 2" (Lines 188-189), I see MDSC cluster (only one, not clusters) in healthy as well. Nevertheless, the authors later studied the correlation in neutrophil clusters (Lines 134-135) without any reason given.

We appreciate your understanding. Regarding the statement about MDSC clusters, we apologize for the error in mentioning neutrophils instead. We have corrected it to MDSC in the revised manuscript. We also acknowledge that MDSC cluster(s) were observed in healthy samples, albeit accounting for a smaller proportion (2 percent) of the population, as you mentioned. This information has been included in the manuscript to provide a comprehensive understanding of the MDSC distribution across different sample types. Thank you for pointing out these issues, and we have made the necessary revisions to improve the clarity and accuracy of the manuscript.

  • The authors only included up-regulated genes in analysis, excluding down-regulated genes, without any given reason.

Thanks for your consideration. We have added this paragraph in the discussion section:

“Another limitation of our study is that we did not thoroughly investigate the potential role of TE downregulation in gene expression related to COVID-19 infection. Our study observed a general upregulation of TEs, with fewer TEs being downregulated, thus excluding the role of TE downregulation in the gene expression changes observed. However, it is important to explore the possible impact of TE downregulation on gene expression in future studies to fully understand the complex dynamics between TEs and viral infections, particularly in the context of COVID-19.”

Minor concerns:

  • Figure 1, "batch effect correction", not "btach effort correction".

Thank you for pointing out the typo in Figure 1. We apologize for the error and appreciate your feedback. The correct term is "batch effect correction," and we have made the necessary correction to ensure accuracy.

  • Lines 65-66: "bronchoalveolar lavage (BAL) fluid", not BALF, since BALF is not used any more in the context. Using BAL hereafter.

We appreciate your feedback, and we will make the necessary correction in the manuscript to use "BAL" instead of "BALF" consistently throughout the text.

  • Lines 90-93: same line for the same setting. 

We appreciate your feedback, and we will make the necessary correction

  • Line 120: full name of KNN needs to be given.

Thank you for pointing that out. In line 120, the full name of KNN (K-Nearest Neighbors) should be provided for clarity.

  • Lines 138-143: legend for Figure 1, not a paragraph of context.

Thank you for bringing this to our attention.

  • Line 145: "This" should be deleted.

Thank you for bringing this to our attention.

  • ID is missing in most of GO, Wiki, KEGG or reactome analysis. GO should have been noted as CC, MF or BP.

Thank you for your feedback. We have added the ID numbers of the GO, Wiki, KEGG, and Reactome analyses in the supplementary material.

  • While the Discussion part is long, the Results part should have been more detailed for interpretation of figures and tables.

Thank you for your feedback. We have considered your suggestions and made revisions to the Results section, including the figures and captions, to improve readability and provide a more detailed interpretation of the results. These improvements aim to enhance the understanding of the findings for readers. We appreciate your valuable input and believe that the revised manuscript now provides a clearer and more comprehensive presentation of the results.

Round 2

Reviewer 1 Report

Comments and Suggestions for Authors

The authors have addressed the issues.

Author Response

R1: The authors have addressed the issues.

Thank you for your comment; we sincerely appreciate your valuable feedback.

Reviewer 2 Report

Comments and Suggestions for Authors

Figure labels are too small to read.

Algorithms versions are needed.

Author Response

  1. Figure labels are too small to read.

Thank you for notifying us; we have addressed this concern by enlarging the font size in the final version

  1. Algorithms versions are needed.

Thank you for highlighting the importance of including algorithm versions, and we have incorporated the relevant algorithm versions used in our study.